# Business Intelligence Adoption for Small and Medium Enterprises: Conceptual Framework

**Ibrahim Abdusalam Abubaker Alsibhawi *** , **Jamaiah Binti Yahaya** and **Hazura Binti Mohamed**

Research Centre for Software Technology and Management, Faculty of Information Science and Technology, Universiti Kebangsaan Malaysia, Bangi 43600, Selangor, Malaysia
* Correspondence: p99726@siswa.ukm.edu.my

**Abstract:** All businesses have many issues, especially small and medium enterprises trying to survive with traditional technology. Therefore, enterprises need to adopt business intelligence by using the management of information technology systems to overcome the issues. This study proposes a conceptual framework that identifies the potential factors that influence the adoption of business intelligence systems in the SME industry in Libya. Therefore, this study was established based on two main theories: the technology acceptance model (TAM) and the unified theory of adopting and using technology (UTAUT). In line with the previous studies that investigated this type of influence, this study recommended a conceptual framework containing several factors: change management, knowledge sharing, information quality, IT project management, the perceived usefulness of a BIS, and the perceived ease of adoption of a BIS. This study did not consider the environmental factors' effect on adopting a BIS (business intelligence system); this is due to the different characteristics of each small and medium enterprise in terms of the sector or industry type.

**Keywords:** business intelligence; business analytics; information technology performance management; framework; Libya case study

## 1. Introduction

Information systems (ISs) play a very relevant role worldwide in the modern economy, allowing organisations and companies to perform various complicated and straightforward functions at high speeds [1,2]. Thus, over the years, throughout the world, information systems and information technologies have been constantly evolving, causing a substantial impact on the way information is treated today. Information worldwide helps to achieve the objectives of an organisation; it also allows the firm to improve the organisation's decision-making in a productive and competitive way [3]. Suppose we take as an example the companies that have grown worldwide [4]. In that case, it can be concluded that the success of their growth is largely due to good technological management that results in a rapid response or good decision-making in the face of different problems that arise. They arise in the business world, which allows them to have customer satisfaction efficiently and effectively [5].

Most existing companies generate, store, and modify an enormous amount of data about any activity registered in the company through data management applications, which are becoming more complicated to use and obsolete [6]. Due to this need, systems that offered support solutions for decision-making began to appear in the 1980s, which are now known as the term "business intelligence," coined by Howard Dresner of the Gartner Group in 1989 [7]. This term is intended to be the basis for gathering all kinds of technologies capable of extracting corporate data stored by the different management systems and treating them in such a way that, by presenting them to any person or user, they can obtain intellectual knowledge to carry out the necessary tasks for the successful achievement of the proposed goals in the business.

The methodological approach to business intelligence is also known as the "business intelligence model," which has different levels regarding the type and treatment of information. It is based on three actions: processes and activities, management, and strategy, each associated with the operational scorecard, management scorecard, and balanced scorecards. The basic objective is to support organisations sustainably and to continuously improve their competitiveness by providing the necessary information for decision-making. The first to coin the term was Howard Dresner, who, when he was a Gartner consultant, popularised "business intelligence" (BI) as an umbrella term to describe a set of concepts and methods that improve decision-making by using information about what has happened (facts) [8]. According to the Gartner definition, business intelligence is "an interactive process to explore and analyse structured information about an area (normally stored in a Data Warehouse), to discover trends or patterns, from which to derive ideas and draw conclusions" [9].

However, as they are large companies, there are also problems in this regard that they face, such as the lack of capacity to manage data volume and many tools that allow for better decision-making within companies, while large companies suffer from having a greater response capacity before large companies with global influences. SMEs are defined as businesses with a few employees and a small revenue (in Libya, less than 50 employees and less than USD 700,000) [10]. In addition, SMEs suffer from not having a response in the world market; this is where the importance of studying small and medium-sized SMEs is given in this research.

The purpose of this research was to propose a framework that illustrates the issues and challenges associated with the adoption of business intelligence systems in small and medium-sized businesses, as well as to identify the adoption factors for business intelligence systems in small and medium-sized businesses. For this purpose, the first section explains the theoretical framework related to the factors used in this study. While the proposed factors are discussed in a separate section, a subsection is specified for each factor. The proposed relationships are illustrated in a framework diagram. A conclusion is included at the end of this paper.

## 2. Background of Study

The SMEs in Libya are essential for the economic growth and social development of the country, in terms of their contribution to the gross domestic product (GDP) and the generation of jobs, so much so that due to their characteristics, capacities, and internal dynamics, they have become the object of study of some national and international organisations, authors, and researchers in order to describe and solve them.

According to Libyan regulations, SMEs are defined as businesses with fewer than 50 employees and revenues of less than USD 700,000 [10]. According to Shibani [11], SMEs have their own characteristics and dimensions; however, they present occupational and financial limitations established by each country's regulations. Therefore, it can be pointed out that no homogeneous term or universal characteristic allows for a notion or conception about what SMEs are. For its part, the term "SME" refers to small and medium-sized companies. SMEs do not have a defined structure; they are generally constituted only by the owner in the administrative part, who makes all the decisions about prices, hiring, and salaries. SMEs do not have a defined job description, and the hired personnel fulfil various functions or positions [12].

Small and medium sized businesses have different definitions in different countries, but generally, a small business is an organisation with a limited number of employees, and typically, the employees is below fifty. In Libya, the Small Business Administration defines small businesses as firms employing fifty or fewer persons. In comparison, medium enterprises are firms employing fifty to two hundred and fifty persons, as shown in Figure 1.

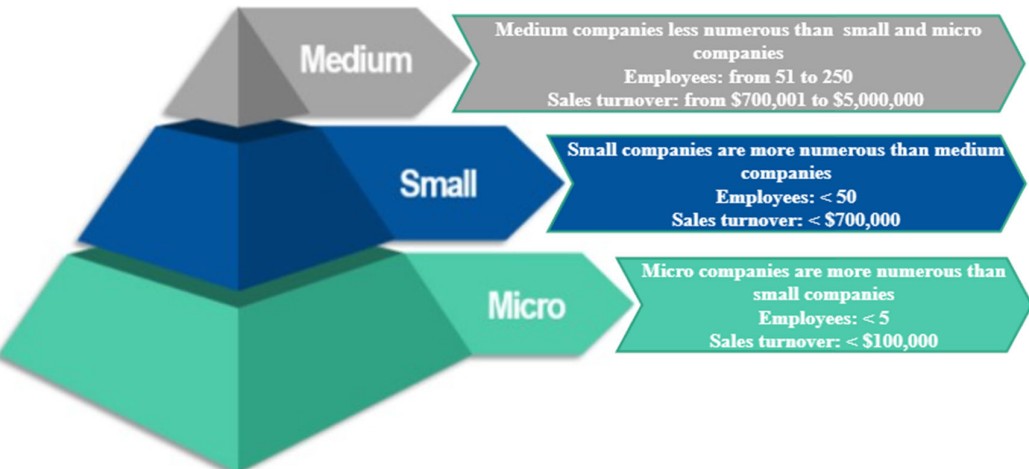

**Figure 1.** SMEs classification in Libya. Source: Ministry of Industry—Libya (2020).

*Background of Small and Medium Enterprises*

Many small and medium sized businesses exist with little adequate knowledge on the part of the business owners about the formulation of business-related growth strategies. These businesses do not succeed very well due to the low skill levels of these owners and business managers [13]. According to Roever, S. [14], the lack of informal small and medium sized business operations policies from the authorities, which will guarantee the right to livelihood for informal business owners and managers, results in failures. Marhaeni, A. A. I. N., N. N. Yuliarmi, and N. D. Setiawina [15] pointed out that small and medium sized businesses can achieve better performance improvements by formalising business operations. These businesses must perform very well because the higher the number of high-performing businesses, the better the economic conditions will be [16].

Boi, K., and V. Dimovski [17] argued that business intelligence can be seen as a process or a product, that considers the intelligence made up of several processes: perception, storage, learning, communication, and decisions. According to Souibgui, M., F. Atigui, S. B. Yahia, and S. S.-S. Cherfi [18], "business intelligence" is defined in terms of gathering, processing, interpreting, and communicating the necessary information in the decision-making processes. Haeckel and Nolan [19] conceptualised it as the ability to capture, share, and extract meaning from market signals. These concepts emphasise (a) the recognition of the value of the information in the external environment of the organisation for the definition of its actions from the management of the information life cycle, which covers the perspectives of the information systems and the context of their repercussions; (b) the informational processes of value addition; and (c) the decision-making process.

The challenges and competitive advantages that can be obtained to achieve success and sustainability in the SME market largely depend on obtaining meaningful information from data and differentiating patterns and trends that allow for more precise decision-making [20]. In the beginning, SMEs lacked computational and technological tools that allowed information to be processed, so most decisions (strategic, tactical, and operational) were made by intuition [21]. Luhn [22], the pioneer of information sciences and a researcher for the company International Business Machines Corp. (IBM), used the term "business intelligence system." Luhn and H. P. [23] referred to an automatic system that accepts information in its original format and disseminates the data to the correct places. To meet these objectives, techniques such as the auto abstraction and auto encoding of documents and the creation and automatic updating of user profiles were used; however, these techniques were mainly statistical and were not effective if there were no facilities in the communication systems and the means of entry and exit. These techniques evolved until the appearance of a set of concepts and methods to improve decision-making in businesses using fact-based support systems, named "BI" (business intelligence) [24]. The

term has significantly evolved and currently includes broad categories of applications, technologies, and methodologies that allow extracting, gathering, classifying, transforming, analysing, and carrying out transactions with structured information through direct exploitation (queries, reports, etc.) or making use of analysis to convert information into knowledge [25].

However, business intelligence not only depends on the use of state-of-the-art methodologies and technologies but is also affected by the lack of knowledge of both internal and external factors (e.g., economic environment, market behaviour, legislation, currency exchange, level of internal communication, internal operations, etc.) involved in an organisation [26]. These factors generally influence all business stakeholders (e.g., customers, competitors, partners, etc.) [27]. Therefore, as a whole, this guides business decisions, allowing the generation of strong competitive advantages.

Caseiro, N., and A. Coelho [28] studied the direct impact of business intelligence on startup business performance as well as the indirect impact through network learning and innovativeness. The results revealed business intelligence's significant and positive impact on innovativeness, network learning, and start-up business performance. Hence, paying attention to the role of business intelligence will improves organisational network learning and performance. Torres, R., A. Sidorova, and M. C. Jones [29] found that business intelligence has a significant and positive impact on organisational performance, as this impact enables firm performance through the business process change. Furthermore, Sua Vugec, D., V. Bosilj Vuki, M. Peji Bach, J. Jakli, and M. Indihar Temberger [30] discovered that fully mediated business intelligence alignment has a significant and positive impact on organisational performance.

End users' use and acceptance of information technologies are additional ways for the system to be successful [31]. Although there are many theories about the use of information technologies by individual consumers, the technology acceptance model (TAM) has been one of the most widely used theories in studies such as those by Verma, S., S. S. Bhattacharyya, and S. Kumar [32]; Mohammad and adwan Moh'd [33]; Sohn, K., and O. Kwon [34]; and Hu, Ding, Li, and Chen [35]; developed by Davis [36]; this model is based on the "planned behaviour theory" of Fishbein, M., and I. Ajzen [37] and aims to understand and explain individuals' acceptance and the use of technological developments. The first model developed was completed in 1989 by Davis, F. D., R. P. Bagozzi, and P. R. Warshaw [38], with some additions [39], and took its place in the literature.

The main purpose of the development of the TAM was to investigate the effects of external variables on internal variables such as belief, attitude, and intention. In this model, the perceived ease of use and perceived usefulness were the two most important factors that explained the use of the system. These two factors help kill the users' perception of the information technology system [40]. In addition to these two main determinants, it is thought that different external factors will significantly affect users' adoption of the system in the TAM. Because the first research on the TAM was generally designed for companies' information systems and applied to professional users and Since individuals are not as knowledgeable as professionals about their use of technology, TAMs used in the first years should be changed to appeal to the end user [41]. For this reason, it is recommended to develop the model by adding new variables to include humanitarian and social factors [42]. These models, which are created by adding new variables to the variables in the TAM model, are called expanded-TAMs [43].

The UTAUT preserves the theoretical foundations of the TAM by combining social impact and environmental factors with the concepts of performance and effort expectation [44]. Venkatesh, Morris, Davis, and Davis [45] stated that performance expectation, effort expectation, social influence, and facilitating conditions play an important role as direct determinants of user acceptance. Compared with the TAM, the UTAUT provides a more comprehensive definition with a 70% acceptance rate in the intention to use. The UTAUT model consists of four structures that affect the user's acceptance and use of information technology. The three direct elements of the intention to use are performance

expectations, effort expectations, and social impact. Intent and facilitating conditions are shown as direct factors of user behaviour. The model states that the attitude towards technology use, self-efficacy, and anxiety are not direct determinants of intention [45].

The two most important determinants of system usage and intention are usefulness and the perceived ease of implementation. Since previous research such as that by Ul-Ain, Vaia, and DeLone [46] and Bach, Zoroja, and Čeljo [47] has indicated that both the perceived usefulness (PU) and perceived ease of use (PEOU) directly impact the business intelligence system (BIS) implementation in general, these relationships are needed in the context of BIS implementation, as stated within the original TAM model. Technology-driven strategy is defined by Gatignon, H., and J.-M. Xuereb [48] as the development, integration, and usage of new technologies in new product and service development. Recent research has shown that a technology-driven strategy has a significant impact on the development of information technology capabilities [49]. Therefore, the direct influence of the technology-driven strategy on BIS's perceived usefulness is tested. Information quality and organisational factors affect user satisfaction with technology and influence beliefs about using it [50,51]. The effect of information quality on the PU and PEOU has been tested by Foshay, N., A. Taylor, and A. Mukherjee [52]; and Kohnke, O., T. R. Wolf, and K. Mueller [53]. They found that information quality influences adoption success by positively impacting the usefulness and perceived ease of implementation.

Amoako-Gyampah and Salam [54] extended the TAM in project communications, training, and shared beliefs. Their findings suggested that training and communication in project communication influenced the TAM since the perceived usefulness and perceived ease of use contribute to behavioural intention to use the technology. This leads to the issues that emerge from the impact of project management on the perceived ease of implementation. Ngai, Law, and Wat [55] suggested successful business process change, user support and involvement, and the vendors' expertise as critical success factors in ERP projects. Markus [56] and Rosati and Lynn [57] suggested that managing organisational change is critical for successful IT project implementation. Al-Zayyat, Al-Khaldi, Tadros, and Al-Edwan [58]; and Bach, Čeljo, and Zoroja [59] concluded that knowledge sharing enables a project team to reduce rework and compresses the time it takes to plan projects. They also stated that having the "right knowledge" available to the "right person(s)" at the "right time" allows for greater control over the project throughout the project's lifecycle by reducing uncertainty. Therefore, project management is driven by change management [60] and knowledge sharing in companies.

According to the TOE framework (technology-organisation-environment), the adoption of technological innovation is influenced by three aspects of a company's context [61]: the organisational context, which is related to resources and their internal characteristics; the context of the environment, within which it carries out its business processes; and the technological context, which is formed by the internal and external technologies related to the organisation that are available in the marketplace.This is a framework for examining the adoption, at the level of organisations (and not individuals), of various ICT information systems [62], products, and services, which are very widespread in ICT adoption and whose advantage is its independence from the size of the company [20], providing a global image of the adoption of technology that predicts the impact on the activities of the value chain and the subsequent diffusion of the factors that influence business decisions.

The use of BIS has received a lot of attention in the literature [63]. The majority of the literature has relied heavily on the TAM, which is regarded as an important theoretical framework for interpreting adoption factors.In this regard, previous studies such as those by Kohnke, Wolf, and Mueller [53]; Foshay, Taylor, and Mukherjee [52]; Hou, [64]; and Boonsiritomachai, McGrath, and Burgess [65] stand essentially on the TAM as listed in Table 1, besides the integration of new factors that treat the TAM weakness. At least four limitations of these studies can be mentioned.

First, the model-based evaluation of management interventions focuses on predicting the use of technologies but not on increasing a user's performance. There is not neces-

sarily a positive relationship between use and performance. As stated by Goodhue, and Thompson [66], a technology may be used, but that does not imply improvements in user performance. Secondly, the limitation of the metadata role in BIS use is related to its ability to predict the actual use of a technology. Turner, Kitchenham, Brereton, Charters, and Budgen [67] addressed the fact that behavioural intent was a good predictor of actual use. Several authors, such as Ain, N., G. Vaia, W. H. DeLone, and M. Waheed [68], and LaBrie, Steinke, Li, Cazier [69], have questioned some results in the metadata role in BIS use based on self-reporting since the limitations that instruments based on the self-perceptions of users have are known.

Third, most of the research has been conducted by measuring the variables of the expectation-confirmation model in relatively homogeneous groups. Indeed, when the model was measured, it was performed with groups of users [70]. This limited the possibility of generalising the results obtained to real environments, which tend to be rather heterogeneous. In other words, it is doubtful whether the expectation-confirmation model could continue to be valid when technology adoption occurs within an organisation where its members exhibit different levels of skills. This is especially true for SMEs with highly diverse organisational environments. In the same sense, it remains to be demonstrated the applicability of the expectation-confirmation model in situations typical of organisational environments, such as the technological adoption of work teams or acceptance of complex technologies. However, the use of relatively homogeneous samples not only calls into question the applicability of the expectation-confirmation model but also its completeness. It can be argued that the model has not included important variables such as information quality. Fourth, a limitation of the business intelligence maturity model that can be mentioned is the model's eminently quantitative nature, which ignores the transformation behaviour towards adoption as its assumption without an intention phase towards adoption.

**Table 1.** The previous framework for BIS adoption.

| Source | Framework | Limitations |
|---|---|---|
| [53] | The model-based evaluation of management interventions | The model-based evaluation of management interventions focuses on predicting the use of technologies but not on addressing the role of IT management resistance towards adoption. It necessary to test this factor as it determines to which extent the IT management supports adoption |
| [52] | The metadata role in BIS use | The limitation of the metadata role in the BIS neglects the effect of the perceived ease of the adoption of the BIS on the perceived usefulness of the BIS, as the perceived usefulness relies on the degree of the ease of adoption. |
| [64] | The expectation-confirmation model | The expectation confirmation model relies on three major factors, namely technological, organisational, and environmental, while the quality of information is out of the categorising of this model. The information quality plays an essential role in producing the optimal output of the BIS, which should be considered. |
| [65] | The business intelligence maturity model | The limitation of the business intelligence maturity model that can be mentioned is the eminently quantitative nature of the studies related to the model, which neglects the transformation behaviour towards doption, as its assumption without an intention phase towards adoption. This phase comprises the measurements of the perceived ease of the adoption and the perceived usefulness of the BIS. |
| [71] | The three-pronged approach model | The limitation of the three-pronged approach model is that it neglects the role of information quality, which plays an essential role in determining the BIS output quality |
| [72] | The integrated theoretical approach model | The limitation of the integrated theoretical approach model is that it neglects the role of information quality, which plays an essential role in determining the BIS output quality |

### 3. The Proposed Conceptual Framework

Regarding the BIS adoption and organisational performance model, Bhatiasevi, V., and M. Naglis [71] presented a model that was based on technology, organisation, and environmental factors. These factors rely on three main determinants that are integrated towards adopting a BIS in SMEs, taking into consideration the integration of the balance scorecard approach to better understand the degree of influence each factor has on the adoption of business intelligence. Based on the UTAUT model, the intention to use and, in turn, the use of technology is determined by four main factors: the performance expectation [73], or the degree to which a person thinks that using the system will help them to improve their performance at work; the expectation of effort, or how easy it is to use the system; the social influence, or the degree to which a person thinks that their social referents will be using the system; and the social norm, or the degree to which a person thinks that the system is The use of the technology is explained directly by the intention of use and the facilitating conditions. In turn, intended use is directly determined by performance expectancy, effort expectancy, and social influence [74]. This theory also includes four moderating variables in the relationship between the four basic variables and the intention and use of technology: gender, age, will, and experience. Venkatesh, Morris, Davis, and Davis [45] reported that the UTAUT explains 70% of the variance in the intention to use. The UTAUT is one of the models that is most often used to study how people adopt and use technology.

Foshay, and Kuziemsky [75] provided a conceptual framework for BIS use in SMEs. The proposed model was constructed based on two phases that lead to adoption. The first phase relies on four dimensions of information quality, which generate attitudes toward data cognition and affect [76]. In the second phase, these attitudes affect the perceived usefulness and the ease of use designed by the technology acceptance model. Hou, [64] assumed three main contexts that led to SMEs' adoption of a BIS: the attitude toward data cognition, attitude toward data cognition affect, and perceived usefulness and ease of use. Technological factors, organisational factors, and environmental factors represent these factors. Integrating these factors leads to evaluating the BIS before its adoption and actual use.

In line with the SMEs' proposed framework for the BIS implementation, Boonsiritomachai, McGrath, and Burgess [65] proposed a conceptual framework that contained nine factors that directly lead to the BIS's adoption by SMEs. These factors can be categorised into three main categories: first, BIS capability factors such as relative advantage, complexity, compatibility, and absorptive capacity; second, business capability factors such as organisational resources, management innovativeness, and management IT knowledge; and third, business external environment factors such as competitive pressure and vendor selection. Based on the above literature and the previous studies, the current study proposed the following conceptual framework, which is shown in Figure 2.

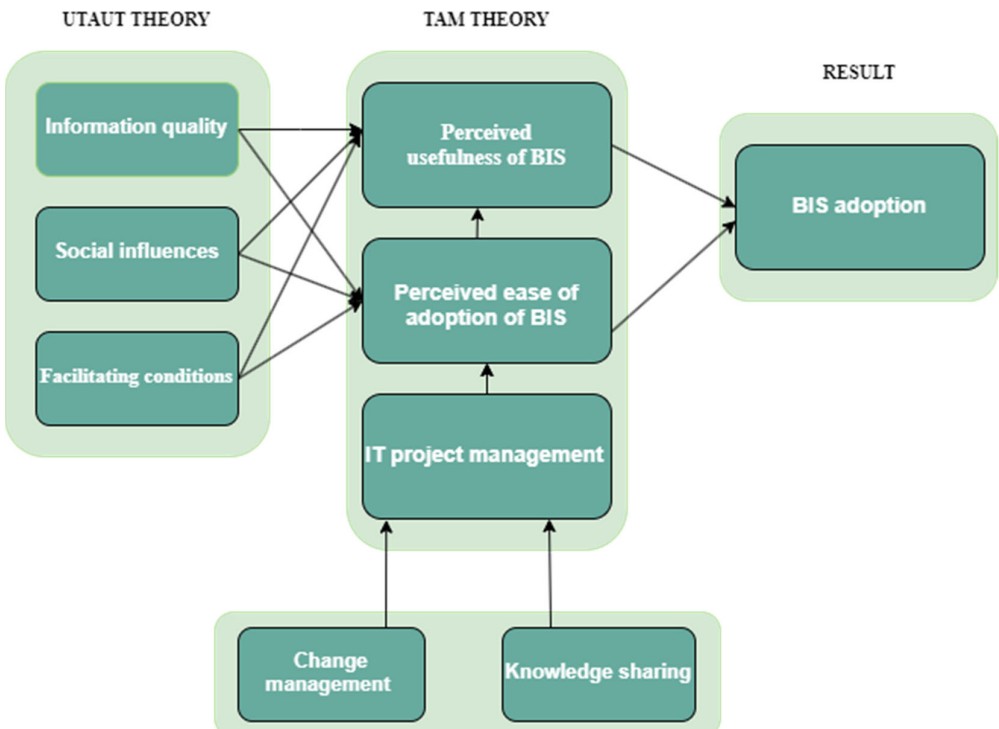

**Figure 2.** Proposed conceptual framework.

## 4. Conclusions

The adoption of a BIS has received much attention in the previous literature. Most of the literature has relied more on the TAM, which is considered an essential theoretical framework that interprets the adoption factors. In this regard, previous studies such as those of Kohnke, Wolf and Mueller [53], Foshay, Taylor and Mukherjee [52], Hou, [64]; and Boonsiritomachai, McGrath and Burgess [65]; stand essentially on the TAM, besides the integration of new factors that treat the TAM weakness. At least four limitations of these studies can be mentioned. SMEs do not recognise the BIS software solutions' competitive advantage compared to other technical solutions because they are typically employed as an integrated model within some bigger company systems. As a result, the factor that has yet to be identified as a risk to the project's success is the perception of the BIS's comparative advantage. Additionally, the requirement for the presentation of its advantages to business employees was seen as one of the causes of effective BIS adoption; therefore, the danger of BIS adoption does not result from a lack of clear and comprehensible presentation of benefits to company employees. The widespread availability of BISs today and their use as an integrated module within another system may account for such an outcome.

The proposed conceptual framework in this study could have several contributions. First, in order to improve the previous literature on BIS adoption, variables that affect BIS acceptability in businesses that have previously deployed and utilised part or all of the modules of integrated information systems should be looked at. The scope of the technology acceptance model should be expanded to include IT project management in businesses, information quality, and technology-driven strategies. With the testing of external variables on features reported in the TAM relying on the UTAUT, this suggested conceptual framework would enhance the usefulness of earlier research employing the TAM technique. Secondly, the current level of BIS utilisation in Libyan SMEs should also be measured, giving data on the implementation's status. The perspective of BIS adoption may have future practical repercussions for technology-driven initiatives, IT project management teams, management methods for implementing BIS in the future, and end users' perspectives on using BIS. In terms of the development of BIS solutions, the

study may also have useful ramifications for the planning and design of BIS solutions in the future, taking into account the key factors that influence their acceptance in companies.

The proposed conceptual framework of this study provides two phases of adoption of the BIS by SMEs, specifically in Libya. The first phase is the fertile ground for adoption, relying on supportive IT management that considers mitigating technology resistance within the organisation, in addition to the mission of technology knowledge diffusion. This phase includes good resources, which include good information quality inputs, social support, and good facility conditions. The effective integration of these factors within the first phase will ease the second phase, which focuses on addressing the potential benefits and ease of using BIS technology.

This study reveals that the effective adoption of a BIS by SMEs should consider several factors related to the business's resources and capabilities. The assessment of these elements reflects the flexibility of the business to effectively adopt the BIS. The potential usefulness and ease of use should be based on the high level of information quality required by the business, as well as the social support and good facility conditions that assist SMEs in smoothly adopting the BIS. Effective IT management plays a significant role in guiding the adoption, as they take care of introducing the BIS to the business, besides providing a high degree of flexibility in executing the BIS adoption at an appropriate phase.

Based on the previous studies that investigated the factors that affect the adoption of a BIS in SMEs, this study proposed a conceptual framework that constructed several factors, which were change in management, knowledge sharing, information quality, IT project management, the perceived usefulness of the BIS, and the perceived ease of the adoption of the BIS to find out their effect on the adoption of business intelligence in the SME industry in Libya. Even though this work adds to the body of research on BIS adoption in small- and medium-sized businesses, it should be emphasised that this study had several restrictions. First, only the sector was included in the study's limited sample size. Second, only two main models were used in this study to propose the conceptual framework: the TAM and the UTAUT. The study should thus be expanded upon or restricted to SMEs that operate in the primary or secondary sectors as part of future research. In order to obtain a deeper understanding of the findings and more specific evidence of the link between the established variables, it is also advised that future studies qualitatively analyse this research.

**Author Contributions:** Drafted the original manuscript, conceptualization, literature analysis, I.A.A.A.; Conceptualization and methodology, J.B.Y.; Investigation and supervision and validation H.B.M. All authors have read and agreed to the published version of the manuscript.

**Funding:** This research was funded by the Malaysian Fundamental Research Grant Scheme (FRGS) under the code 1/2015/ICT04/UKM/02/1.

**Institutional Review Board Statement:** Not applicable.

**Informed Consent Statement:** Not applicable.

**Data Availability Statement:** Not applicable.

**Conflicts of Interest:** The authors declare no conflict of interest. The funding sponsors had no role in the design of the study; in the collection, analyses, or interpretation of data; in the writing of the manuscript: or in the decision to publish the results.

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
