# Peer review of "Business Intelligence Adoption for Small and Medium Enterprises: Conceptual Framework"

_applsci, doi:10.3390/app13074121_

Round 1
Reviewer 1 Report
1.The problem statement is more general and it is only explained about the sensor nodes energy consumption. Include the problem scenario with detailed explanation where need to be concentrated.
2. The abstract need to more precise and include the percentage of improvement in the values compared with the previous values in the comparison .
3.Expand BIS before use in the first time.
4. what is out come of The Proposed Conceptual Framework?
5. What research contribute done on this paper?
6.Figure 1 nothing is represented ? figure 2 need to improve.
Author Response
Dear reviewers,
We appreciate your precious time in reviewing our paper and providing valuable comments. Your helpful and insightful words led to possible improvements in the current version. The authors have carefully considered the statements and tried our best to address every one of them. The authors welcome further constructive comments if any. We hope the manuscript, after careful revisions, meets your high standards.
Below we provide the point-by-point responses. All modifications in the manuscript have been highlighted it.

Reviewer 2 Report
- The English writing of the manuscript needs improvement. Therefore, it could benefit greatly from professional editing to improve technical writing and English.
- Please mention your study limits and suggest some future research topics
- In References, the sources are written in different styles. Please update the reference list. It is necessary to bring in accordance with the requirements of the journal for the design of References. If possible, indicate DOI.
- Please use some innovative keywords.
- Please mention your study limits in the abstract.
- The Conclusions should reflect what the practical application of the results obtained in this study is. In what climatic conditions should the recommendations of the authors be taken into account?
Author Response

(The authors gave the same response as above.)

Round 2
Reviewer 1 Report
author addressed all comments
Reviewer 2 Report
accept